# Effect of 5-Aminolevulinic Acid (5-ALA) in “ALADENT” Gel Formulation and Photodynamic Therapy (PDT) against Human Oral and Pancreatic Cancers

**DOI:** 10.3390/biomedicines12061316

**Published:** 2024-06-13

**Authors:** Domenica Lucia D’Antonio, Simona Marchetti, Pamela Pignatelli, Samia Umme, Domenico De Bellis, Paola Lanuti, Adriano Piattelli, Maria Cristina Curia

**Affiliations:** 1Department of Medical, Oral and Biotechnological Sciences, “Gabriele d’Annunzio” University of Chieti-Pescara, 66100 Chieti, Italy; domenica.dantonio@unich.it (D.L.D.); simona.marchetti@phd.unich.it (S.M.); ummesamia95@gmail.com (S.U.); 2Villa Serena Foundation for Research, Via Leonardo Petruzzi 42, 65013 Città Sant’Angelo, Italy; 3COMDINAV DUE, Nave Cavour, Italian Navy, Stazione Navale Mar Grande, Viale Jonio, 74122 Taranto, Italy; pamelapignatelli89p@gmail.com; 4Center for Advanced Studies and Technology (CAST), “Gabriele d’Annunzio” University of Chieti-Pescara, 66100 Chieti, Italy; domenico.debellis@unich.it (D.D.B.); p.lanuti@unich.it (P.L.); 5Department of Medicine and Aging Sciences, “Gabriele d’Annunzio” University of Chieti-Pescara, 66100 Chieti, Italy; 6School of Dentistry, Saint Camillus International University of Health and Medical Sciences, Via di Sant’Alessandro 8, 00131 Rome, Italy; apiattelli51@gmail.com; 7Facultad de Medicina, UCAM Universidad Católica San Antonio de Murcia, 30107 Murcia, Spain

**Keywords:** 5-aminolevulinic acid, photodynamic therapy, pancreatic cancer, oral cancer, protoporphyrin, reactive oxygen species, CAL-27, CAPAN-2, apoptosis, cell cycle

## Abstract

Oral squamous-cell and pancreatic carcinomas are aggressive cancers with a poor outcome. Photodynamic therapy (PDT) consists of the use of photosensitizer-induced cell and tissue damage that is activated by exposure to visible light. PDT selectively acts on cancer cells, which have an accumulation of photosensitizer superior to that of the normal surrounding tissues. 5-aminolevulinic acid (5-ALA) induces the production of protoporphyrin IX (PpIX), an endogenous photosensitizer activated in PDT. This study aimed to test the effect of a new gel containing 5% *v*/*v* 5-ALA (ALAD-PDT) on human oral CAL-27 and pancreatic CAPAN-2 cancer cell lines. The cell lines were incubated in low concentrations of ALAD-PDT (0.05%, 0.10%, 0.20%, 0.40%, 0.75%, 1.0%) for 4 h or 8 h, and then irradiated for 7 min with 630 nm RED light. The cytotoxic effects of ALAD-PDT were measured using the MTS assay. Apoptosis, cell cycle, and ROS assays were performed using flow cytometry. PpIX accumulation was measured using a spectrofluorometer after 10 min and 24 and 48 h of treatment. The viability was extremely reduced at all concentrations, at 4 h for CAPAN-2 and at 8 h for CAL-27. ALAD-PDT induced marked apoptosis rates in both oral and pancreatic cancer cells. Elevated ROS production and appreciable levels of PpIX were detected in both cell lines. The use of ALA-PDT as a topical or intralesional therapy would permit the use of very low doses to achieve effective results and minimize side effects. ALAD-PDT has the potential to play a significant role in complex oral and pancreatic anticancer therapies.

## 1. Introduction

Pancreatic cancer (PC) and oral squamous-cell carcinoma (OSCC) are both lethal conditions with a poor outcome and an increasing incidence. In 2021, pancreatic and oral cancers accounted, ively, for 62,210 and 54,010 new cancer cases. Survival for both cancers is poor, with pancreatic cancer being the seventh-leading cause of cancer mortality worldwide, while oral cancer accounts for 1.8 percent of all cancer deaths [1,2]. Age-adjusted death rates of oral and pancreatic cancer have been rising, on average, by 0.4% and 0.5%, respectively, each year over 2010–2019 [3,4]. The average survival time for pancreatic cancer is low, partly because an early diagnosis is difficult. At present, surgical resection is the only potential cure for pancreatic cancer; however, only 20% of patients can proceed with surgical removal of the tumor [5] and rates of recurrence are high [6]. The oral cancer treatments available during recent decades have remained unchanged [7]. Among these, surgical resection, with or without adjuvant therapy, is a commonly used treatment. However, oral cancer is very aggressive and there is a high risk of developing second primary cancers. Unresectable oral cancers are treated with palliative systemic therapy and/or palliative radiotherapy [8] and with checkpoint inhibitors, acting on the tumor microenvironment [9].

Photodynamic therapy (PDT), whose general principle was first described in 1900, is an effective treatment for both precancerous and malignant conditions [10]. It consists of the use of a photosensitizer capable of inducing cell and tissue damage when activated by exposure to low-level visible light [11]. Several photosensitizers are activated by different total light doses, dose rates, and specific wavelengths of light radiation [11]. PDT selectively acts on cancer cells, which have an accumulation of photosensitizer superior to that of the normal surrounding tissues. In contrast to ionizing radiation, PDT is generally safer than other treatments for the surrounding normal tissues due to its cell selectivity and lack of cumulative toxicity, and it can be safely applied to previously irradiated tissues [12]. The introduction of 5-ALA in the Ppix metabolic pathway, regulated by the feedback control resulting from the presence of cellular concentrations of heme, leads to the intracellular accumulation of Ppix [13]. 

5-aminolevulinic acid (5-ALA) is metabolized to protoporphyrin IX, the final intermediate in the heme biosynthetic pathway, and an endogenous fluorescent photosensitizer. 5-ALA has been used to formulate a solution for treating potentially malignant lesions of the oral cavity by applying it topically through a saturated gauze placed against the injury or through a customized temporary dental prosthesis [14]. The application of 5-ALA in some lesions has proved difficult due to dilution by saliva and the difficulty of maintaining positioning. For this reason, it has been administered by intralesional injection or in a temperature-dependent gel-state formulation to optimize the bioavailability of the drug. A special sol–gel formula is marketed as ALADENT, which contains 5% of 5-ALA. It is composed of a mixture of poloxamers that ensure the stability of the active ingredient. This liquid thermolabile formulation transforms into gel at body temperature (>28 °C) [15,16]. The gel form allows better mucoadhesion, facilitating topical administration, greater hydrophilicity, relative solubility in water and lipids, and constant ionization [17]. This formulation showed a greater release of 5-ALA and, therefore, greater effectiveness in the accumulation of protoporphyrin in comparison with conventional cream and ointment vehicles, in the cure of dermatological and gastrointestinal disorders [18,19,20,21]. ALADENT is also used in the treatment of periodontitis and peri-implantitis thanks to its antibacterial activity against Gram-positive and -negative bacteria, such as Porphyromonas gingivalis, Enterococcus faecalis, Escherichia coli, Staphylococcus aureus, and Veillonella parvula, with or without irradiation. However, the strongest antibacterial effect was achieved with 25 min of 50% ALAD incubation followed by 5 min of a red LED [22].

PpIX, produced by 5-ALA metabolism, selectively kills neoplastic cells either directly, by stimulating the production of reactive oxygen species (ROS), or indirectly, by damaging tumor vascularity and activating immune responses against cancer cells [23]. The cytotoxic effect of PDT is due to cellular necrosis or apoptosis, depending on whether the 5-ALA localizes in lysosomes or cell membranes or penetrates mitochondria during light emission [24]. Cells with high rates of metabolic activity, such as cancer cells, inflammatory cells, and bacteria have lower amounts of ferrochelatase (FECH)*,* an enzyme capable of metabolizing PpIX, so the production of photoactive porphyrins is more pronounced than in healthy cells [25]. Reduced FECH expression is present in various tumors such as kidney [26], bladder [27], and colorectal [28]. Low FECH activity of cancer cells causes accumulation of PpIX and other porphyrins in tumors, while normal or high FECH activity results in low accumulation of PpIX in tissues [29]. Thus, 5-ALA-based PDT can be a potential therapeutic strategy for the treatment of different tumors. In preclinical studies the anti-tumor efficacy of 5-ALA-mediated PDT has been observed in different cancer types such as skin cancer [30], esophageal cancer [31], colon cancer [28], human glioma [32], breast cancer [33], bladder cancer [34], and hepatocellular carcinoma [35]. 

A lot of illness, like head and neck [36], bladder [37], and Barrett’s esophagus [38] cancers, as well as gynecological neoplasia [39] and actinic keratosis [40] are treated with 5-ALA-PDT. 5-ALA-PDT does not produce the same effects on all types of cancer cells; in fact, it can induce non-inflammatory cell death such as necrosis and apoptosis or inflammatory cell death. This therapy’s use is greatly limited by the lack of knowledge of its mechanisms of action [41]. Other limitations to clinical application are the variability in 5-ALA uptake from cancer cells, the penetration of only 5 mm of 635 nm activating light, and poor oxygen recovery that limits the production of cytotoxic ROS [42]. As regards PC, in a previous paper, 5-ALA at higher doses followed by PDT prolonged animal survival [43].

ALAD intralesional or topical administration with in situ gel formation would allow the use of very low doses below the maximum dose allowed and used in humans, reducing systemic effects and resulting in effective 5-ALA concentrations. The use of ALAD-PDT could allow targeted anticancer therapy to be to carried out, with the possibility of repeating the administration over time with limited side effects. This new formulation of 5-ALA-mediated PDT was used for the first time on squamous oral and pancreatic cancer cells. The objective of this research was to examine how ALADENT affects oral and pancreatic cancer cells by measuring the level of PpIX, oxidative stress, apoptosis, and cell vitality. The aim was also to evaluate cell-cycle modulation after ALAD-PDT. 

## 2. Materials and Methods

### 2.1. Chemicals

ALAD is a gel containing 5% *v*/*v* (4.6 M) of 5-ALA, commercialized as ALADENT by ALPHA Strumenti s.r.l. (Melzo, MI, Italy), and it is covered by European patent EP3727334 with the following description: “pharmaceutical preparation comprising a topically released active ingredient and a heat-sensitive carrier, method of obtaining same, and use of same in the treatment of skin and mucosal infections”. 

The full text of the patent is available at https://register.epo.org/espacenet/regviewer?AP=18836409&CY=EP&LG=en&DB=REG. It is a pharmaceutical preparation comprising an active ingredient 5-ALA, and a heat-sensitive carrier (thermogel) made of 19% poloxamer (PL) P407 and 4% poloxamer P188, with the balance being composed of water. Poloxamer hydrogels have a reversible thermo-responsive sol-to-gel transition (28 °C for ALADENT) which induces the formation of a solid gel from a solution, as a result of their self-assembling into micelles. ALAD remains stable when stored at 4° for up to a week, but after that, its effectiveness decreases. Poloxamer use is approved by the FDA and listed as pharmaceutical excipient in the United States and European Pharmacopoeias; the lack of toxic crosslinking agents renders it more likely than thermogels to show intrinsic biocompatibility as an injectable in situ hydrogel. The presence of a poloxamer mixture in the formulation facilitates the 5-ALA to rapidly enter into the mucous membranes of the target cells, acting as a transmucosal delivery.

### 2.2. Cell Lines and Culture

The investigations were performed on human tongue squamous carcinoma cell line CAL-27 and human pancreatic ductal adenocarcinoma cell line CAPAN-2. HGF-1 cells are a major constituent of the oral microenvironment, in particular of gingival connective tissue, which is the first tissue invaded by oral cancer. So they were used as a non-cancerous control cell line. 

CAL-27 (ATCC-CRL-2095) and CAPAN-2 (CLS, Eppelheim, Germany) cancer cell lines were kindly provided by Dr. Maria Carmela Di Marcantonio and by Dr. Serena Veschi, respectively. HGF-1 cells (ATCC-CRL-2014) were purchased from LGC Standards S.r.l., Sesto San Giovanni, Milano, Italy.

CAL-27 and HGF-1 cells were cultured in Dulbecco’s Modified Eagle’s Medium (DMEM; EuroClone S.p.A., Pero, MI, Italy) and CAPAN-2 cells were cultured in a Roswell Park Memorial Institute medium (RPMI 1640, Corning, Glendale, AZ, USA). All the media were supplemented with 10% fetal bovine serum (FBS), 2 mM L-glutamine, and antibiotics (100 units/mL penicillin and 100 µg/mL streptomycin) (all from EuroClone).

Cultured cells were maintained in 5% CO_2_ at 37 °C and 95% humidity and examined periodically under an inverted phase-contrast microscope. The cells grew as an adhering monolayer and the culture medium was changed once or twice a week. When cells reached 70–80% confluence, they were sub-cultured or seeded.

### 2.3. Light Source and Irradiation Parameters 

An aluminium gallium arsenide (AlGaAs) power LED device (TL-01) characterized by 630 nm ± 10 nm full width half maximum (FHWM) nm wavelength was used as light source (Alphastrumenti, Melzo, MI, Italy). The hand-piece was constituted by 1 LED with 6 mm diameter at the exit and a surface irradiance of 380 mW/cm^2^. During the experiments, the LED hand-piece was mounted perpendicularly to the wells at 0.5 mm of distance with a particular polystyrene box to maintain a constant distance from the light source, and also to obtain a uniform LED irradiation of all the samples. At these conditions, the exit irradiance surface was 380 mW/cm^2^ and the total specific dose was 23 J/cm^2^ for each minute of irradiation. The irradiation was performed under a laminar flow hood in the dark under aseptic conditions in all the experiments.

### 2.4. Cell Treatment

Twenty-four hours after the seeding, the CAL-27, CAPAN-2, and HGF-1 cells were incubated with increasing concentrations of ALAD: 0.05% *v*/*v*, 0.1% *v*/*v*, 0.2% *v*/*v*, 0.40% *v*/*v*, 0.75% *v*/*v*, and 1% *v*/*v*, corresponding, as reported in literature [44,45], to 5-ALA concentrations of 0.23 mM, 0.46 mM, 0.92 mM, 1.84 mM, 3.45 mM, and 4.6 mM, respectively, for different experimental times (2–8 h) in serum-free medium at 37 °C and 5% CO_2_. Then, the cells were exposed to PDT using RED light (630 nm) (ALPHA Strumenti s.r.l) with an intensity of 380 mW/cm^2^ for 7 min with a light dose of 23 J/cm^2^. Subsequently, the serum-free medium was replaced with a medium containing 10% FBS. The control group consisted of untreated CAL-27 and CAPAN-2 (without ALAD) cells. The effects of ALAD-PDT on cell viability, apoptosis, the cell cycle, and ROS levels were assessed at selected concentrations and different time points, as indicated in the related paragraphs.

### 2.5. Cytotoxicity and Cell-Viability Assay (MTS Assay)

Cells in the logarithmic growth phase were seeded onto 96-well plates for 24 h, at different densities according to cell types: 8 × 10^3^ cells/well for CAPAN-2, 10 × 10^3^ cells/well for CAL-27, and 6 × 10^3^ cells/well for HGF-1. Then, cell lines were treated with ALAD-PDT, as reported above. 

After 24 h, MTS solution (10 μL) (Promega’s CellTiter 96^®^ AQueous Non-Radioactive Cell Proliferation Assay, Madison, WI, USA) was added to each well, followed by incubation at 37 °C, 5% CO_2_ for 1 h. All conditions were performed as quintuplicate. The absorbance was determined at 490 nm using a microplate reader (Synergy H1 Hybrid BioTek Instruments). Cell viability was reported as a percentage as compared with the untreated cells (CTRL), recognized as 100%. 

### 2.6. Flow Cytometry Analysis 

#### 2.6.1. Detection of Apoptosis

To discriminate apoptotic cells, propidium iodide (PI) (BD Biosciences, Milano, Italy), which is able to identify damaged membranes, and Annexin V-FITC (fluorescein-isothiocyanate) (BD Biosciences), which allows the detection of phosphatidylserine expression on cell membranes, were used to stain 3 × 10^5^ cells (15 min at RT, 25 °C in the dark) [46]. The use of Annexin V to stain the phosphatidylserine flipped on the external layer of the plasma membrane provides a much more comprehensive picture of the apoptotic process than the identification of the sub-G1 peak [47,48]. Therefore, the AnnexinV/PI analysis was applied to study the ALAD-PDT-induced apoptosis.

Samples were then analyzed by flow cytometry using a FACSVerse cytometer (BD Biosciences). Finally, data were analyzed using FlowJo v10.10.0 software (BD Biosciences, Milano, Italy).

#### 2.6.2. Cell-Cycle Analysis

An amount of 5 × 10^5^ cells per sample were harvested, fixed in 70% cold ethanol and stored at 4 °C. Samples were then resuspended in 50 μg/mL PI (Sigma, St. Louis, MO, USA) and 120 μg/mL RNAse (Sigma, St. Louis, MO, USA), as previously described [49]. Samples were acquired using a FACScanto II flow cytometer (BD Biosciences). Data were analyzed with FlowJo software v10.10.0 (BD Biosciences).

#### 2.6.3. Reactive Oxygen Species (ROS) Levels

Reactive oxygen species production was carried out by staining the cells (5 × 10^5^ cells/sample) with 5 μM of 5-(and-6)-chloromethyl-2′,7′-dichlorodihydrofluorescein diacetate, acetyl ester (CM-H2DCFDA Molecular Probes, Invitrogen, Life-Sciences-Division, Milan, Italy) for 1 h at 37 °C. When ROS are produced, CM-H2DCFDA is oxidized and an increase in green fluorescence is detectable by flow cytometry. Samples were analyzed by using a FACSVerse cytometer (BD Biosciences, San Jose, CA, USA). Data were analyzed using FlowJo v10.10.0 (BD Biosciences).

### 2.7. PpIX Fluorescence Measurements

PpIX intracellular content was determined after ALAD-PDT treatment (4 h for CAPAN-2 and 8 h for CAL-27) at different time points, namely T1 (10 min after the treatment), T2 (24 h after the treatment), and T3 (48 h after the treatment).

An amount of 8 × 10^3^ cells/well for CAPAN-2 and 10 × 10^3^ cells/well for CAL-27 were plated onto 96-well plates and subjected to the ALAD-PDT treatment. Then, cells were treated with a solution of perchloric acid (HClO4) in methanol, as previously described [50], and PpIX fluorescence was measured using a microplate spectrofluorometer (Synergy H1 Hybrid BioTek Instruments, Winooski, VT, USA) at λex/em 405/608. 

### 2.8. Statistical Analysis

Data are expressed as the means ± standard deviation (SD). Statistical analyses were performed using the *t*-test to compare treated to control samples. Differences between groups were assessed with one-way analysis of variance (ANOVA). A *p*-value ≤ 0.05 was considered as significant. Statistical analyses and descriptive statistics were carried out using GraphPad Prism version 9.0 (GraphPad Software Inc., La Jolla, CA, USA).

## 3. Results

### 3.1. Cytotoxicity

The CAL-27 and CAPAN-2 human cancer cell lines were incubated at low concentrations of ALAD (0.05%, 0.10%, 0.20%, 0.40%, 0.75%, 1%) at different times and then exposed to 630 nm RED light irradiation. At first, ALAD-PDT-treated cells were compared with ALAD-treated cells without LED irradiation (Appendix A). Results showed marked differences with the use of light and the study continued only using the ALAD-PDT treatment. Furthermore, the results confirmed literature data on the use of PDT.

After ALAD-PDT, for CAPAN-2 cells, the optimum inhibition efficiencies were obtained at 4 h, and for CAL-27 cells, at 8 h. Very low concentrations of ALD-PDT blocked cell growth and caused apoptosis in both cell lines (Figure 1). The differences were in the extent of cytotoxicity and in the treatment time—4 h for CAPAN-2 cells and 8 h for CAL-27 cells. At the lowest concentration of 0.05% of ALADENT, there was a notable decrease in vitality in both cell lines. For this reason, subsequent analyzes were performed on the two lower concentrations, 0.05% and 0.1%, which corresponded to the “minimum effective dose” (MED).

For CAL-27, the viability was reduced by approximately 70% at all concentrations after 8 h (Figure 1). CAPAN-2 cells showed a higher sensitivity to ALAD-PDT than CAL-27 cells and already, after 4 h incubation, an almost total reduction (80%) in viability was visible (Figure 1) at all concentrations. 

The MTS assay was also performed on human gingival fibroblast (HGF-1) cells under the same conditions to evaluate the selective toxicity of ALAD-PDT on normal cells (Figure 1). The results showed a percentage of cell viability of not less than approximately 50% for both experimental times (4 and 8 h) after treatment. 

### 3.2. Apoptosis Rates and Cell-Cycle Analysis

The influence of ALAD-PDT on the induction of apoptotic cell death was studied by evaluating exposure to phosphatidylserine through Annexin-V staining of CAL-27 and CAPAN-2 cells. For CAPAN-2, given the high cell mortality after 4 h of treatment, the treatment time was reduced to 2 h, maintaining low ALAD treatment concentrations. Figure 2 shows the gating strategy used to identify Annexin-V-positive cells. Briefly, debris was excluded on an FSC-A/SSC-A dot plot and the four sub-populations identified by PI and Annexin V staining were gated. In Figure 3, the fold-change values of the percentage of total Annexin-V-positive cells, calculated with respect to untreated samples, were reported as bars both for 0.05% and 0.1% ALAD-PDT treatments (CAPAN-2: *p*-value = 0.3284, 95% confidence interval −35.9669 to 84.0339; CAL-27: *p*-value = 0.0186, 95% confidence interval 19.4935 to 76.9067). As shown, the induction of apoptosis was increased in ALAD-PDT-treated CAL-27 and CAPAN-2 cells with respect to untreated samples. This effect was dose-dependent with a significant increase in apoptosis between 0.05% and 0.1% ALAD-PDT treatments in CAL-27 cells. Such an increase was also detected in CAPAN-2 cells but without reaching statistical significance.

Cell-cycle analysis was also carried out (Figure 4). As shown, the ALAD-PD treatment did not impact the cell cycle at either at 0.05% or 0.1%.

### 3.3. ROS-Level Variation

ALAD is enzymatically converted into PpIX, that is photoactivated by light and produces ROS, inducing cell death. The fluorescence measured by the DCFH2-DA assay was used to study the ROS production. The ratio between the DCFH2-DA fluorescence of 0.05% or 0.1% ALAD-PDT-treated cells and that of untreated samples, is reported as bars in Figure 5. As shown, the ROS production was increased in ALAD-PDT-treated CAL-27 and CAPAN-2 cells with respect to untreated samples. This effect was dose-dependent with a significant increase in ROS production between 0.05% and 0.1% ALAD-PDT treatments in CAL-27 cells (CAPAN-2: *p*-value 0.0676, 95% confidence interval −0.271360 to 3.29536; CAL-27: *p*-value 0.0034, 95% confidence interval 0.762759 to 1.27399). Such an increase was also detected in CAPAN-2 cells but without reaching statistical significance (Figure 5). 

These results demonstrate that the production of reactive oxygen species was stimulated by photoactivation. 

### 3.4. Generation of Intracellular PpIX

PpIX accumulation analysis was also carried out. PpIX fluorescence was measured in CAPAN-2 and CAL-27 cell lines showing an increase in PpIX that occurred within 10 min (T1) after the treatment time (4 h for CAPAN-2 and 8 h for CAL-27) with ALAD-PDT. The PpIX content was maintained over time (T2, 24 h; T3, 48 h), with slight increases and decreases based on the concentrations of ALAD (0.05% and 0.1%) and on cell type (Appendix A). Data showed statistically significant differences as compared with the control. 

## 4. Discussion 

OSCC and PC are characterized by an increase in incidence worldwide and they are associated with a high mortality rate. This demands the development of new strategies to treat these deadly malignancies. PDT is an alternative therapy that relies on the activation of a light-sensitive photosensitizer using visible light in the presence of molecular oxygen, causing ROS production [51]. The process then leads to irreversible cell damage and localized cell death [52,53]. Numerous advantages, including minimal toxicity to normal tissues, negligible systemic effects, protection of organ function, and high efficacy, make PDT a medical technique for the treatment of cancer patients [54]. The photosensitizer holds the major role in anti-tumor activity [44]. In cells, 5-ALA is converted to protoporphyrin IX (PpIX), a photosensitizer and direct precursor of hemoglobin in the heme synthesis pathway, which accumulates in mitochondria, preferentially in the tumor. This occurs as a result of the reduced activity of ferrochelatase and the lower availability of iron [55]. Considering that the absorbance range of 5-ALA is between 420 and 760, in vitro, we chose a wavelength of 630 nm to observe a high PDT effect. 5-ALA has an anti-inflammatory and immunoregulatory action reducing the expression of Toll-like receptor (TLR) 2 and 4 [56].

PDT is usually defined as a powerful inducer of tumor-cell apoptosis [33]. The rate of apoptosis increased after ALAD-PDT, a sign that its administration causes an arrest in the proliferation of both cell lines but at different times. In fact, for CAPAN-2 cells, the incubation time was reduced due to the high mortality of cells post ALAD-PDT. Another important role in apoptotic cell death is the formation of mitochondrial ROS after ALAD-PDT. It is well known that mitochondrial ROS formation plays a critical role in PDT-induced apoptosis. Mitochondrial inner-membrane permeabilization leads to mitochondrial swelling, cytochrome c release to the cytosol, and activation of the caspase cascade [57]. On the other hand, when singlet oxygen and lipid peroxide induced by ALAD-PDT destroy the plasma membrane, the cell becomes necrotic. It has been reported that the cytosol localization of PpIX caused cell necrosis, while photosensitizer localizing in mitochondria and lysosomes led to more pronounced apoptosis [58].

Mitochondrial ROS production can be of enormous importance for cell proliferation and apoptosis [34,35]. Recently, mitochondria-targeting therapeutic strategies have been considered as a novel approach for cancer treatment. As reported in literature, ALA-induced PpIX is initially localized in the mitochondria [36]. Excessive mitochondrial ROS production may lead to cell death induced by oxidative stress; however, these responses differ in different cell types [37]. Our results demonstrated that ALAD-PDT influenced ROS production in CAPAN-2 cells more than in CAL-27 cells. Thus, we assume that induced mitochondrial ROS production probably provides a clue to the differential susceptibility of CAL-27 and CAPAN cells to ALAD-PDT.

ROS levels were increased in both treated cell lines with respect to untreated samples. The increase was marked in CAPAN-2 cells at both ALAD concentrations. On the other hand, CAL-27 cells showed a significant increase in both apoptosis and ROS production between 0.05% and 0.1% ALAD concentrations, indicating 0.1% as the most effective treatment. These observations confirm literature data on esophageal cancer cells during PDT in mice [59]. 

It has been shown that lower 5-ALA concentrations (1–4 mM) allowed an inhibitory effect after 2–16 h incubation in oral and pancreatic cancer cell lines [44,45].The formulation of 5% 5-ALA with poloxamer mixture facilitates uptake from target cells, reducing the minimum inhibitory concentration of 5-ALA compared with previous concentrations used for PDT. Using a drug at low concentration reduces possible side effects. 

This is the first in vitro study investigating the differences in sensitivity to 5-ALA-PDT between OSCC and PC. Interestingly, CAL-27 and CAPAN-2 cells responded similarly to ALAD-PDT treatment even if ALAD-PDT showed different effects on the two cell lines. In fact, CAL-27 cells were more resistant to treatment, reaching a relevant cytotoxic effect after only 8 h of treatment. These findings are consistent with the pathobiological behavior of OSCC cells, which have a higher metabolic rate and proliferate faster [30,31]. This is probably also due to the greater ability of the cells of the oral cavity to resist mechanical and thermal stress, due to their anatomical location. Confirming this, in the study by Rosin et al., the *SCC9* cell line, isolated from the tongue of a patient with squamous-cell carcinoma, survived after 5-ALA-PDT administration by reducing PpIX synthesis and initiating signaling pathways related to cell proliferation and apoptosis [60]. 

On the contrary, the CAPAN-2 cell line showed a greater response to the treatment after 2 h, the first incubation time tested. Little is known about the in vitro effect of 5-ALA-PDT on human PC cell lines; Khaled et al. assessed the anti-cancer effects of 5-ALA-PDT only when it was combined with oncolytic viral therapy [45].

Preclinical and clinical studies of 5-ALA-PDT against oral and pancreatic cancer showed positive and promising results. In the study by Zhu et al. of oral squamous carcinoma, 5-ALA-mediated PDT both in vitro and in vivo triggered the generation of intracellular ROS with significant cytotoxicity and apoptosis-induction effects, thus inhibiting tumor growth [61]. Furthermore, in the study by Wang et al., topical 5-ALA-PDT proved to be a safe treatment in OSCC patients with locally advanced sites, in addition to platinum-based induction chemotherapy (ICT) [62]. Despite the short observation period and small sample size, the results suggest prospective studies to evaluate the efficacy and safety of topical 5-ALA-PDT as an adjuvant to ICT followed by surgery.

As regards PC treatment, in the study by Regula et al., 5-ALA was administered systemically (intravenously or orally) to animals, with higher doses prolonging animal survival [47]. In another study, the prodrug 5-ALA was selectively activated by endogenous Cathepsin E within PC cells in vitro and in vivo [63]. This new combined approach reduced the viability of pancreatic cancer cells, with few side effects, while sparing normal cells.

ALADENT intralesional or topical administration with in situ gel formation would allow the use of very low doses, below the maximum dose allowed and used in humans, reducing systemic effects and resulting in effective 5-ALA concentrations.

### Limitations and Future Research Directions

This was a preliminary and in vitro study. Further studies to identify the molecular signaling pathways involved in the cell death mechanisms in 5-ALAD-PDT-exposed cancer are required. Only a few functions and possible mechanisms of 5-ALAD-PDT have been elucidated in this research. A better understanding of the pathophysiological characteristics of oral and PC cells is essential for exploring the mechanism of 5-ALA-PDT. Additionally, further studies should be devoted to determining how other related molecules and signaling pathways interact with each other in the process.

Therefore, additional investigations are needed to translate 5-ALAD-PDT findings on alternative therapeutic strategies. Finally, future studies should focus on in vivo functions and mechanisms, which may provide better evidence to facilitate our understanding of the potential clinical implications of 5-ALAD-PDT, considering that various cell lines respond in different ways to this treatment.

## 5. Conclusions

Following the results obtained from experiments, we can confirm that ALAD-PDT markedly inhibited the viability and proliferation of PC and OSCC cell lines through mechanisms involving accumulation of PpIX that, under irradiation, can induce profound ROS production and phototoxicity. When cells were treated with low concentrations of ALAD-PDT, the intracellular PpIX level increased producing a large amount ROS. The results of the study indicate that administered 5-ALA could be successfully interconverted to PpIX in cells, inducing focal cancer-cell death. The absorption of the active substance would be facilitated by this new gel formulation, which would make treatment of OSCC-affected regions, such as the tongue and lingual floor, more convenient and non-invasive. Also, ALAD intralesional administration would allow its application in pancreatic-cancer treatment. By performing localized treatment of the pathological area in the early stages, it would be possible to avoid more invasive surgical and chemotherapy treatments. The initial treatment of oral cancers with ALAD-PDT would reduce the extent and contain the pathology as adjuvant therapy.

PDT and ALAD together slowed the growth of PC and OSCC cells, presenting itself as a promising treatment strategy for oral and pancreatic cancer. Notably, the time responses of the CAL-27 and CAPAN-2 cells were different, probably due to differences in cell type and tumor location. Also, for pancreatic cancer, there may be promising prospects for conservative therapies following studies that can provide clarification. Indeed, several in vitro and in vivo studies need to be performed to better understand the potential of ALAD-PDT as a targeted therapy for cancer. In our research, we have only explored and addressed a few of the possible mechanisms of ALAD-PDT.

These findings indicate that ALAD-PDT may be the breakthrough in anti-cancer adjuvant treatment.

## Figures and Tables

**Figure 1 biomedicines-12-01316-f001:**
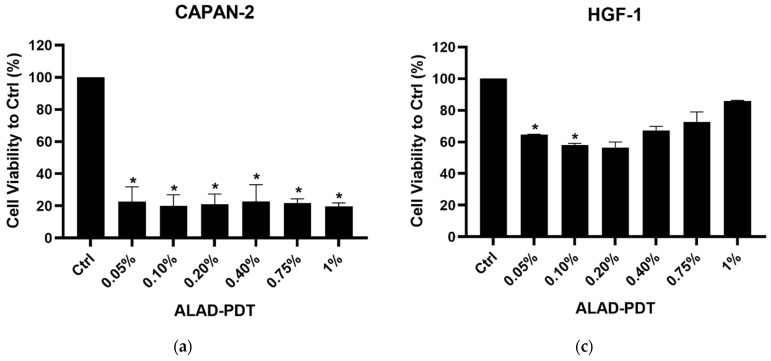
MTS assay after ALAD-PDT treatment. Cell viability of CAPAN-2 cells after 4 h (**a**) and of CAL-27 cells after 8 h (**b**) of treatment. Cell viability of HGF-1 cells after 4 h (**c**) and 8 h (**d**) of treatment. Data shown are means ± standard deviation (SD) of two independent experiments with quintuplicate determinations. * Statistically significant differences as compared with control (* *p* < 0.05; *** *p* < 0.001).

**Figure 2 biomedicines-12-01316-f002:**
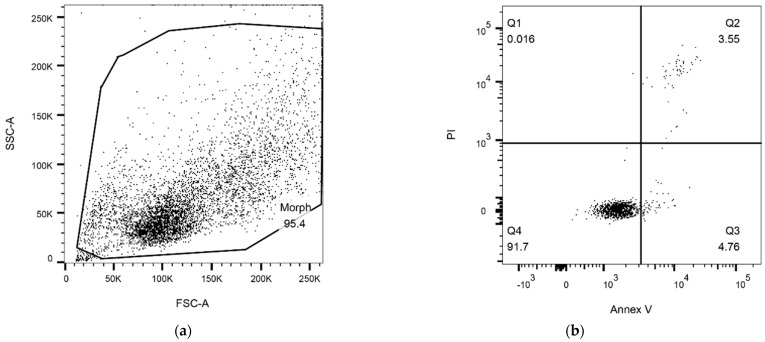
Gating strategy for apoptosis detection: (**a**) All events were gated (morph region) on a Forward Scatter Area (FSC-A)/Side Scatter Area (SSC-A) dot plot. (**b**) Cells were analyzed on a dot plot of Annexin V (Annex V)/propidium iodide (PI). The identified regions are as follows: Q1 = necrotic cells (Annex V−/PI+), Q2 = late apoptotic cells (Annex V+/PI+), Q3 = early apoptotic cells (Annex V+/PI−), and Q4 = live cells (Annex V−/PI−). The represented gating strategy refers to the CAPAN-2 cell line analyzed under logarithmic growth conditions. The same gating strategy was used to analyze all samples.

**Figure 3 biomedicines-12-01316-f003:**
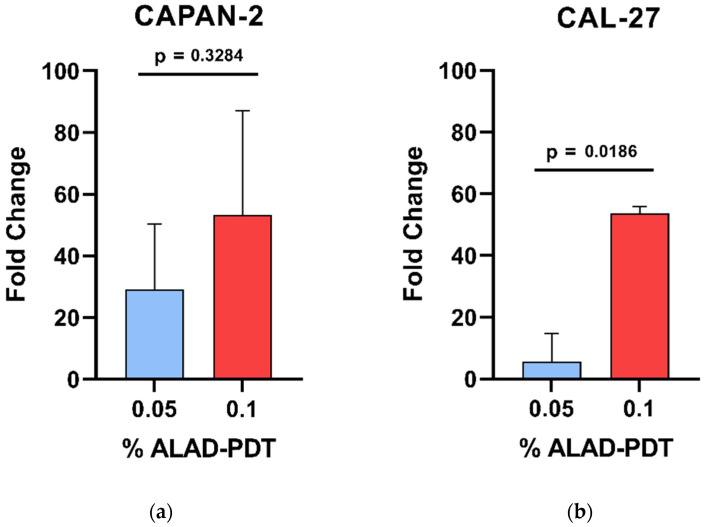
Flow cytometry analysis of apoptosis induced by ALAD-PDT treatment. Bars represent the percentage of fold increase in cells staining positive to Annexin V after cell treatment with two different concentrations, 0.05% (blue bars) and 0.1% (red bars), of ALAD (ALAD-PDT), with respect to the untreated sample set at 1 (CTRL) and, therefore, not represented, both for CAPAN-2 (**a**) and CAL-27 (**b**) cell lines. Percentage of fold-change values were calculated as the ratio between the percentage of Annexin-V-positive cells induced by each treatment (0.05% or 0.1%) and the percentage of Annexin-V-positive cells detected in the respective untreated sample (CTRL). Data are presented as the means ± SD of duplicate experiments. *p* < 0.05 indicates statistical significance.

**Figure 4 biomedicines-12-01316-f004:**
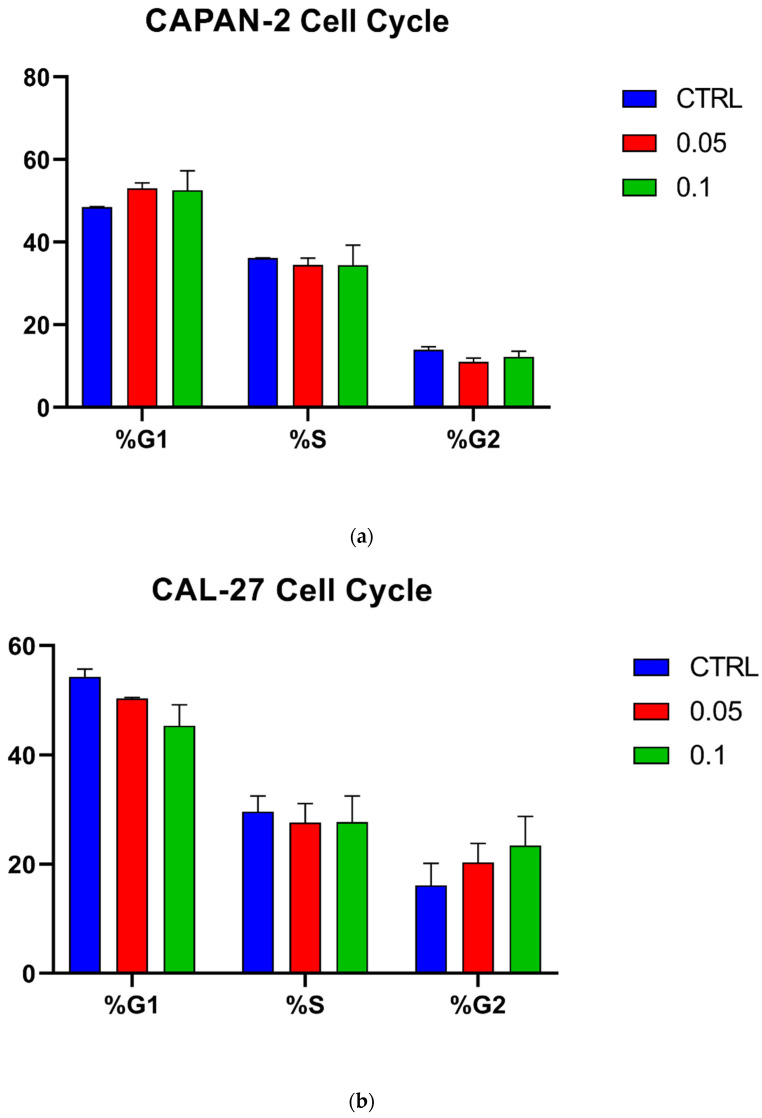
Effect of ALAD-PDT on CAPAN-2 (**a**) and CAL-27 (**b**) cell cycle. Bars represent the percentages (y-axis) of cell distribution throughout the cell-cycle progression (G1, S, and G2 phases) in controls (CTRL, blue bars) and after cell treatment with two different concentrations, 0.05% (red bars) and 0.1% (green bars), of ALAD (x-axis). Data were analyzed using FlowJo v10.10.0 software and are presented as the means ± SD of duplicate experiments.

**Figure 5 biomedicines-12-01316-f005:**
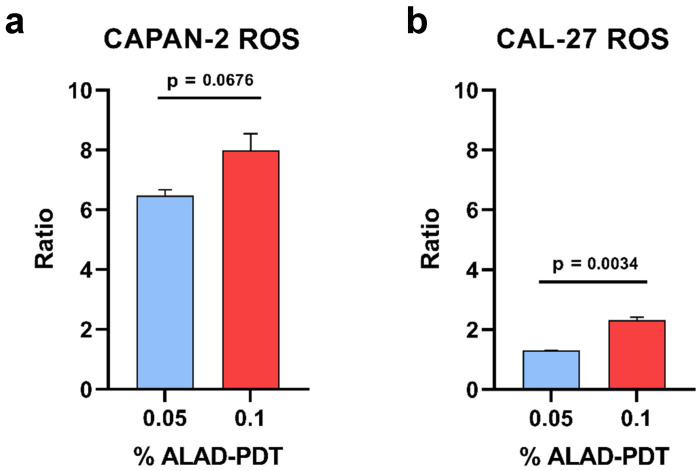
The effect of ALAD-PDT on ROS production by flow cytometry. Bars represent the ratio of ROS production after cell treatment with two different concentrations of ALAD (ALAD-PDT), 0.05% (blue bars) and 0.1% (red bars), and the untreated samples set at 1 (CTRL) and, therefore, not represented, both for CAPAN-2 (**a**) and CAL-27 (**b**) cell lines. Ratio values were calculated as the ratio between the mean fluorescence intensity (MFI) of each treatment (0.05% or 0.1%) and the MFI of the untreated sample (CTRL). Data are presented as the means ± SD of duplicate experiments. *p* < 0.05 indicates statistical significance.

## Data Availability

Data is contained within the article or Appendix A.

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
