# Peer review of "Effect of 5-Aminolevulinic Acid (5-ALA) in “ALADENT” Gel Formulation and Photodynamic Therapy (PDT) against Human Oral and Pancreatic Cancers"

_biomedicines, 2024, doi:10.3390/biomedicines12061316_

Round 1
Reviewer 1 Report
Comments and Suggestions for Authors
Overall, the authors have improved the manuscript based on the comments in the previous review. Many issues have been fixed by adding more figures, such as investigating the intracellular Ppix enrichment. However, the supplementary figures are not included in this submission. Meanwhile, some problems are still unsettled, and the manuscript should be improved further for publication.
1. The abstract is still disappointing and should be reorganized and improved because the current version is full of format issues, missing emphasis, and some sentences are confusing and ambiguous. For example, "ALAD-PDT on human oral CAL-27 and pancreatic CAPAN-2 cancer cell lines"; "ALADENT (ALAD) is a gel containing 5% v/v 5-ALA capable of easily penetrating cell membranes", whether the gel or 5-ALA is penetrable? A comma is missing for the sentence in the abstract: "PpIX accumulation was also measured"; "4h" and "6h" should be "4 h" and "6 h". These formatting issues are ridiculous for a 3rd round revision.
2. "Protoporphyrin IX, Ppix" is inconsistently written in the manuscript. Some place is "Ppix" while some place is "PpIX".
3. I can't find the supplementary figures related to this resubmission to biomedicine.
4. The gel preparation has been included in the manuscript, but I did not find any characterization of the materials. Electron microscope images are usually required for hydrogel characterization.
5. I'm afraid I have to disagree with the explanation for the missing controls in the ROS and cell apoptosis assays. I am pretty sure these controls were included in similar assays in abundant PDT literature.
Author Response
Reviewer 1
Overall, the authors have improved the manuscript based on the comments in the previous review. Many issues have been fixed by adding more figures, such as investigating the intracellular Ppix enrichment. However, the supplementary figures are not included in this submission. Meanwhile, some problems are still unsettled, and the manuscript should be improved further for publication.
- The abstract is still disappointing and should be reorganized and improved because the current version is full of format issues, missing emphasis, and some sentences are confusing and ambiguous. For example, "ALAD-PDT on human oral CAL-27 and pancreatic CAPAN-2 cancer cell lines"; "ALADENT (ALAD) is a gel containing 5% v/v 5-ALA capable of easily penetrating cell membranes", whether the gel or 5-ALA is penetrable? A comma is missing for the sentence in the abstract: "PpIX accumulation was also measured"; "4h" and "6h" should be "4 h" and "6 h". These formatting issues are ridiculous for a 3rd round revision.
- "Protoporphyrin IX, Ppix" is inconsistently written in the manuscript. Some place is "Ppix" while some place is "PpIX".
- I can't find the supplementary figures related to this resubmission to biomedicine.
We thank the reviewer for his valuable suggestions. Following the reviewer's suggestions we reorganized and improved the abstract, we corrected "Protoporphyrin IX in "PpIX" and we added the supplementary figures at the end of the manuscript.
- The gel preparation has been included in the manuscript, but I did not find any characterization of the materials. Electron microscope images are usually required for hydrogel characterization.
The gel containing 5% v/v of 5-ALA (ALADENT) used in the study is a pharmaceutical preparation already marketed and not produced in our laboratories. The chemical-physical characterization has been indicated in the Materials and Methods section. However, for more in-depth details you can consult the link https://register.epo.org/espacenet/regviewer?AP=18836409&CY=EP&LG=en&DB=REG..
We modified the paragraph from 2.1 Chemicals in the Material and Methods section.
“It is a pharmaceutical preparation comprising an active ingredient 5-ALA, and a heat-sensitive carrier (thermogel) made of 19 % Poloxamer (PL) P407 and 4 % Poloxamer P188, the balance being composed of water. Poloxamer hydrogels have a reversible thermo-responsive sol-to-gel transition (28 °C for ALADENT ) which induces the formation of a solid gel from a solution, as a result of their self-assembling in micelles. ALAD remains stable when stored at 4 degrees for up to a week, but after that, its effectiveness decreases.”
Noteworthy other papers in which the same product was used have been published in MDPI (Gels and others)
- I'm afraid I have to disagree with the explanation for the missing controls in the ROS and cell apoptosis assays. I am pretty sure these controls were included in similar assays in abundant PDT literature.
We thank the reviewer for underlying this point, both for apoptosis and ROS assays untreated controls were acquired and in both cases were used to normalize data related to the treatments. In detail in Figure 3 we calculated the percentage of fold change values as the ratio between the percentage of Annexin V positive cells induced by each treatment (0.05 % or 0.1%) and the percentage of Annexin V positive cells detected in the respective untreated sample (CTRL). In Figure 5, ratio values were calculated as the ratio between the mean fluorescence intensity (MFI) of each treatment (0.05 % or 0.1%) and the MFI of the untreated sample (CTRL).
Reviewer 2 Report
Comments and Suggestions for Authors
The article provides an insightful exploration of the effects of 5-ALA in the ALADENT gel formulation and its application in PDT against oral and pancreatic cancer cell lines. The study is thorough and covers multiple aspects of the experimental setup, results, and implications. However, several areas require clarification, elaboration, and restructuring to enhance the manuscript's clarity and impact.
Introduction:
· The introduction provides a good background but can be streamlined to focus more on the novelty and significance of the study.
· Add recent statistics and more references on the current limitations of PDT and 5-ALA in cancer treatments.
· Clearly state the hypothesis and objectives at the end of the introduction.
Materials and Methods:
· Provide more details on the preparation and stability of the ALADENT gel. Include specific storage conditions and shelf-life data.
· Clarify the rationale behind the chosen concentrations of ALADENT. Were these based on preliminary studies or literature?
· The cell line culture conditions and treatment protocols are adequately described. However, the method of ensuring consistency in light exposure across all samples needs more detail.
· Explain the choice of control groups more clearly, particularly the use of HGF-1 cells as non-cancerous controls.
· The statistical methods section should specify the software used for analysis and include the justification for the chosen statistical tests.
Results:
· Results are well-presented but could benefit from more detailed statistical analysis. Include confidence intervals and p-values for all comparisons.
· Figures and tables should be self-explanatory. Ensure all axes, labels, and legends are clear and consistent.
· The apoptosis and ROS results are interesting but need more in-depth discussion on the mechanisms. Explain why different incubation times were used for CAL-27 and CAPAN-2 cells.
· Consider adding a section on the potential clinical implications of the findings in the results discussion.
Discussion:
· The discussion should better connect the findings to existing literature. Compare and contrast with other studies more explicitly.
· Address potential limitations of the study, such as the in vitro nature of the experiments and the need for in vivo validation.
· Provide a clearer statement on the translational potential of ALADENT-PDT. What are the next steps towards clinical application?
Conclusion:
· The conclusion is succinct but should reiterate the key findings and their significance. Emphasize the novelty and potential impact on clinical practice.
· Suggest specific future research directions based on the study’s findings.
Comments on the Quality of English LanguageNeeds minor editing
Author Response
Reviewer 2
The article provides an insightful exploration of the effects of 5-ALA in the ALADENT gel formulation and its application in PDT against oral and pancreatic cancer cell lines. The study is thorough and covers multiple aspects of the experimental setup, results, and implications. However, several areas require clarification, elaboration, and restructuring to enhance the manuscript's clarity and impact.
Introduction:
- The introduction provides a good background but can be streamlined to focus more on the novelty and significance of the study.
- Add recent statistics and more references on the current limitations of PDT and 5-ALA in cancer treatments.
- Clearly state the hypothesis and objectives at the end of the introduction.
We thank the reviewer for his valuable suggestions. Following the reviewer's suggestions we implemented the Introduction section by providing more references on the current limitations of PDT and 5-ALA in cancer treatments, by stating the hypothesis and objectives and by emphasizing the novelty and significance of our study.
We highlighted how this new 5% 5-ALA gel formulation activated by PDT was used for the first time on cancer cells.
5-ALA in the past years has had a limitation in terms of bioavailability and delivery. ALADENT uses a new delivery, a mix of poloxamers, which should improve the bioavailability of the active ingredient at the site of action.
Materials and Methods:
- Provide more details on the preparation and stability of the ALADENT gel. Include specific storage conditions and shelf-life data.
We added in the paper information regarding the stability, storage mode, composition, and lifespan of the photosensitizing hydrogel used.
- Clarify the rationale behind the chosen concentrations of ALADENT. Were these based on preliminary studies or literature?
The chosen concentration of ALADENT were based on literature.
We added more details in Materials and Methods section:
2.4. Cell treatment
“24 h after the seeding, the CAL-27, CAPAN-2 and HGF-1 cells were incubated with increasing concentrations of ALADENT: 0.05% v/v, 0.1% v/v, 0.2% v/v, 0.40% v/v, 0.75% v/v, 1% v/v, corresponding, as reported in literature [44,45] to 5-ALA concentration of 0.23 mM, 0.46 mM, 0.92 mM, 1.84 mM, 3.45 mM and 4.6 mM, respectively, for different experimental times (2-8 h) in serum-free medium at 37 °C and 5% CO2”.
- The cell line culture conditions and treatment protocols are adequately described. However, the method of ensuring consistency in light exposure across all samples needs more detail.
We added more details in Materials and Methods section:
2.3. Light Source and Irradiation Parameters
“During the experiments, the LED hand-piece was mounted perpendicularly to the wells at 0.5 mm of distance with a particular polystyrene box to maintain a constant distance from light source and also to obtain a uniform LED irradiation of all the samples”.
- Explain the choice of control groups more clearly, particularly the use of HGF-1 cells as non-cancerous controls.
We explained the choice of HGF-1 cells as non-cancerous controls and added the following sentence in Materials and Methods section, 2.2. Cell lines and culture:
“HGF-1 are a major constituent of oral microenvironment, in particular of gingival connective tissue, that is the first tissue invaded by oral cancer. So they were used as a non-cancerous control cell line.”
- The statistical methods section should specify the software used for analysis and include the justification for the chosen statistical tests.
The statistical method section has been implemented as required
Results:
- Results are well-presented but could benefit from more detailed statistical analysis. Include confidence intervals and p-values for all comparisons.
As suggested, confidence intervals and p-values for all comparisons were added
- Figures and tables should be self-explanatory. Ensure all axes, labels, and legends are clear and consistent.
As suggested, all axes, labels and legends were reviewed.
- The apoptosis and ROS results are interesting but need more in-depth discussion on the mechanisms.
Explain why different incubation times were used for CAL-27 and CAPAN-2 cells.
We wrote in RESULTS section:
3.1. Cytotoxicity
“After ALAD-PDT, for Capan-2 cells the optimum inhibition efficiencies were obtained at 4 h and for CAL-27 at 8 h”
- Consider adding a section on the potential clinical implications of the findings in the results discussion.
See below
Discussion:
- The discussion should better connect the findings to existing literature. Compare and contrast with other studies more explicitly.
Following the reviewer's suggestions we modified the Discussion section
- Address potential limitations of the study, such as the in vitro nature of the experiments and the need for in vivo validation.
See below
- Provide a clearer statement on the translational potential of ALADENT-PDT. What are the next steps towards clinical application?
See below
Conclusion:
- The conclusion is succinct but should reiterate the key findings and their significance. Emphasize the novelty and potential impact on clinical practice.
We highlighted in the Conclusion how the effect of the increase in intracellular ROS and apoptosis produced by ALAD-PDT can be exploited to reduce the size of the tumor before surgical therapies
- Suggest specific future research directions based on the study’s findings.
We added a new paragraph entitled LIMITATIONS and FUTURE RESEARCH DIRECTIONS in DISCUSSION to respond to the following points suggested by the reviewer:
- Consider adding a section on the potential clinical implications of the findings in the results discussion.
- Address potential limitations of the study, such as the in vitro nature of the experiments and the need for in vivo validation.
- Provide a clearer statement on the translational potential of ALADENT-PDT. What are the next steps towards clinical application?
- Suggest specific future research directions based on the study’s findings
“Limitations and future research directions
This is a preliminary and in vitro study. Further studies to identify the molecular signaling pathways involved in the cell death mechanisms in 5-ALAD-PDT-exposed cancer is required. Only a few functions and possible mechanisms of 5-ALAD-PDT have been elucidated in this research. A better understanding of the pathophysiological characteristics of oral and PC cells is essential for exploring the mechanism of 5-ALA-PDT. Additionally, further studies should be devoted to determining how other related molecules and signaling pathways interact with each other in the process.
Therefore, additional investigations are need to translate 5-ALAD-PDT findings on alternative therapeutic strategies. Finally, future studies should focus on in vivo functions and mechanisms, which may provide better evidence to facilitate our understanding of the potential clinical implications of 5-ALAD-PDT, considering that various cell lines respond in different ways to this treatment.”
Round 2
Reviewer 1 Report
Comments and Suggestions for Authors
The authors have improved the manuscript accordingly, and I have no more concerns.
Minor typos and formatting problems should be noted and corrected:
1. "mW/cm2" and "J/cm2" should be corrected as "380 mW/cm2" and "J/cm2" in section 2.3.
2. Figure 1 and the supporting figures should be numbered as a, b, c, d...
3. Numbers (a, b) are missing in Figure 2.
Author Response
June 10, 2024
Reviewer 1 second round
We thank the reviewer and we corrected the minor typos and formatting problems
Reviewer 2 Report
Comments and Suggestions for Authors
Thank you for your responses
Author Response
June 10, 2024
We thank the reviewer